# Caspase-1 Regulates the Apoptosis and Pyroptosis Induced by Phthalocyanine Zinc-Mediated Photodynamic Therapy in Breast Cancer MCF-7 Cells

**DOI:** 10.3390/molecules28165934

**Published:** 2023-08-08

**Authors:** Chunjie Ma, Yu Wang, Wei Chen, Ting Hou, Honglian Zhang, Hongguang Zhang, Xu Yao, Chunhui Xia

**Affiliations:** 1Pharmacy Department, Qiqihar Medical University, Qiqihar 161006, China; 2020220126@stu.qmu.edu.cn (C.M.); httxsx@163.com (T.H.); zhanghonglian_2006@163.com (H.Z.); zhanghg@qmu.edu.cn (H.Z.); yaoxu@qmu.edu.cn (X.Y.); 2Basic Medicine Department, Qiqihar Medical University, Qiqihar 161006, China; wyfr1970@126.com; 3College of Chemistry and Chemical Engineering, Qiqihar University, Qiqihar 161000, China; chenwei150080@163.com

**Keywords:** TαPcZn-PDT, ROS, apoptosis, pyroptosis, siRNA-caspase-1

## Abstract

Photodynamic therapy (PDT) is an innovative and perspective antineoplastic therapy. Tetra-α-(4-carboxyphenoxy) phthalocyanine zinc (TαPcZn)-mediated PDT (TαPcZn-PDT) has shown antitumor activity in some tumor cells, but the manner in which caspase-1 is involved in the regulation of apoptosis and pyroptosis in the TαPcZn-PDT-treated breast cancer MCF-7 cells is unclear. Therefore, effects of TαPcZn-PDT on cytotoxicity, cell viability, apoptosis, pyroptosis, cellular reactive oxygen species (ROS), mitochondrial membrane potential (ΔΨm), caspase-1, caspase-3, and nuclear transcription factor-κB (NFκB) in MCF-7 cells was firstly examined in the present study. The findings demonstrated that TαPcZn-PDT resulted in the increase in cytotoxicity and the percentage of apoptotic and pyroptotic cells, the reduction in cell viability and ΔΨm, the production of ROS and the activation of caspase-1, caspase-3 and NFκB in MCF-7 cells. Furthermore, the results also revealed that siRNA-targeting caspase-1 (siRNA-caspase-1) attenuated the effect of TαPcZn-PDT on apoptosis, pyroptosis and the activation of caspase-1, caspase-3 and NFκB in MCF-7 cells. Taken together, we conclude that caspase-1 regulates the apoptosis and pyroptosis induced by TαPcZn-PDT in MCF-7 cells.

## 1. Introduction

Cancer is one of the major diseases affecting human health and is the leading cause of global death. At present, breast cancer is one of the most prevalent malignant tumors in women and is treated mainly with surgery, radiation and chemotherapy. These treatments often have poor therapeutic effects, a high risk of postoperative recurrence, pain and other side effects, which make them unable to better improve patients’ postoperative quality of life as well as their physical and mental health, and no longer meet medicine-based needs [1,2].

Photodynamic therapy (PDT) is a new type of treatment for unresectable malignant tumors or those with more complications after resection, which is selective for target tissues and can reduce the damage to normal tissues in domestic and international studies [3,4]. Under the excitation of specific wavelengths, the photosensitizer molecules switch from the ground state to the excited state and interact with the ground state oxygen in the surrounding tissues to produce reactive oxygen species (ROS), which in turn kills tumor cells [5,6]. Some photosensitizers for antitumor therapy (temoporfin, talaporfin, verteporfin and 5-aminolevulinic acid) have been marketed, but they have certain defects, such as short absorption wavelengths, limited tissue penetration and insufficient targetability [7,8,9]. Phthalocyanine is a planar macrocyclic conjugated system consisting of four isoindole units (Figure 1) [10]. Phthalocyanine zinc photosensitizers, which have stable physical and chemical properties, strong absorption in the wavelength range (600~800 nm), low phototoxicity and good photodynamic properties, are one of the most highly potential antitumor photosensitizer candidates [11,12,13,14].

Apoptosis (type I programmed cell death) and pyroptosis (novel type of programmed cell death different from apoptosis) are two cellular processes that play a crucial role during the development and maintenance of tissue homeostasis [15,16,17,18,19]. Accumulating evidence has demonstrated that apoptosis and pyroptosis are two key events in the treatment process of anticancer drugs [20,21]. Caspase-1, a member of the caspase family of proteases, is a key protease in pyroptosis [16]. Accumulating evidence has demonstrated that caspase-1 plays a crucial regulatory role in apoptosis and pyroptosis induced by PDT in cancer cells [22,23,24,25].

Our group has previously proven that tetra-α-(4-carboxyphenoxy) phthalocyanine zinc (TαPcZn)-PDT could obviously induce apoptosis in hepatocellular carcinoma Bel-7402 cells and LoVo colon carcinoma cells [26,27]. Nevertheless, how TαPcZn-PDT induces apoptosis and pyroptosis in human breast cancer MCF-7 cells is not clear. The aim of this study was to investigate the effect of caspase-1 on apoptosis and pyroptosis induced by TαPcZn-PDT in MCF-7 cells. The results demonstrated that caspase-1 regulates the apoptosis and pyroptosis induced by TαPcZn-PDT in MCF-7 cells.

## 2. Results

### 2.1. TαPcZn-PDT Produces Cytotoxicity in Breast Cancer MCF-7 Cells

The effect of TαPcZn-PDT on cytotoxicity in MCF-7 cells was revealed by Calcein AM/PI assay using fluorescence microscopy. Compared with the control treatment, TαPcZn-PDT gradually decreased green fluorescence and increased red fluorescence, pointing to the fact that TαPcZn-PDT produces the cytotoxicity of MCF-7 cells in a dose-dependent pattern (Figure 2).

### 2.2. TαPcZn-PDT Reduces Cell Viability of MCF-7 Cells and Has Little Effect on Mammary Epithelial MCF-10A Cells

The effect of TαPcZn-PDT on viability in MCF-7 cells was detected by MTT assay using microscopy and multi-function reader. Compared with the control treatment, TαPcZn-PDT gradually decreased blue-purple crystals and cell viability in MCF-7 cells, indicating that TαPcZn-PDT inhibits the proliferation of MCF-7 cells in a dose-dependent pattern (Figure 3). In addition, the results also showed that TαPcZn-PDT had little effect on MCF-10A cells, indicating that TαPcZn might be an effective and safe photosensitizer for breast cancer (Figure 3).

### 2.3. ROS and ∆Ψm Are Involved in Apoptosis and Pyroptosis Induced by TαPcZn-PDT in MCF-7 Cells

To evaluate whether TαPcZn-PDT induced apoptosis and pyroptosis in MCF-7 cells, the apoptosis and pyroptosis was firstly quantitated by a flow cytometry analysis of Annexin V/PI double-stained cells. Compared with the control treatment, TαPcZn-PDT led to an increase in the percentage of apoptotic and pyroptotic cells in a dose-dependent pattern (Figure 4A).

To further investigate apoptosis and pyroptosis induced by TαPcZn-PDT in MCF-7 cells, the levels of factors and proteins associated with apoptosis and pyroptosis, including caspase-3, caspase-1 and NFκB, was detected by Western blot. Compared with the control treatment, TαPcZn-PDT activated caspase-3, caspase-1 and NFκB in a dose-dependent pattern (Figure 4B).

To evaluate whether ROS is involved in the TαPcZn-PDT-treated MCF-7 cells, the ROS levels were detected by a DCFH-DA/Hoechst 33342 assay of fluorescence microscope. Compared with the control treatment, TαPcZn-PDT gradually resulted in an increase in the ratio of green to blue fluorescence intensity, indicating that TαPcZn-PDT induced ROS production in a dose-dependent pattern (Figure 5).

To evaluate whether mitochondrion is involved in the TαPcZn-PDT-treated MCF-7 cells, the ∆Ψm was detected by JC-1/Hoechst 33342 assay of fluorescence microscope. Compared with the control treatment, TαPcZn-PDT gradually decreased red fluorescence and increased green fluorescence in the cytoplasm, indicating that TαPcZn-PDT induced a decrease in ∆Ψm in MCF-7 cells in a dose-dependent pattern (Figure 6).

### 2.4. Caspase-1 Modulates Apoptosis and Pyroptosis Induced by TαPcZn-PDT in MCF-7 Cells

#### 2.4.1. siRNA Silencing of Caspase-1

To detected the silencing effect of caspase-1 in the TαPcZn-PDT-treated MCF-7 cells, the cells were transfected with caspase-1-1 (forward 5′-CCACUGAAAGAGUGACUUUTT-3′, reverse 5′-AAAGUCACUCUCUUCAGUGGTG-3′) and caspase-1-2 (forward 5′-GGAAGACACUCAUUGAACAUATT-3′, reverse 5′-UAUGUUCAAUGAGUCUUCCAA-3′). Compared with the siRNA-caspase-1-2 treatment, there is a higher transfection rate of approximately 90% in the siRNA-caspase-1-1 treatment (Figure 7(A1,A2,B1,B2)). After the transfection of siRNA-caspase-1-1 and siRNA-caspase-1-2, the expression levels of caspase-1-1 is significantly less than that of caspase-1-2, indicating that the silencing effect of siRNA-caspase-1-1 is stronger than that of siRNA-caspase-1-2 (Figure 7(A3,B3)). Therefore, this experiment was followed by the selection of siRNA-caspase-1-1.

#### 2.4.2. siRNA-caspase-1 Inhibits Apoptosis and Pyroptosis Induced by TαPcZn-PDT in MCF-7 Cells

To test whether caspase-1 modulates apoptosis and pyroptosis induced by TαPcZn-PDT in MCF-7 cells, the effect of siRNA-caspase-1 on apoptosis and pyroptosis was investigated in the present study. Compared with the TαPcZn-PDT treatment, siRNA-caspase-1 decreased the percentage of apoptotic and pyroptotic cells (Figure 8A), and attenuated the activation of caspase-3, caspase-1 and NFκB (Figure 8B), indicating that caspase-1 regulates apoptosis and pyroptosis induced by TαPcZn-PDT in MCF-7 cells.

## 3. Discussion

Phthalocyanine-PDT exhibited anti-tumor activity in cancer cells [11,12,13,14]. Our previous study has shown that TαPcZn-PDT can apparently inhibit the proliferation of hepatocellular carcinoma Bel-7402 cells [27]. The present study firstly investigated the significant inhibitory effects of TαPcZn-PDT on MCF-7 cells and MCF-10A cells. Our findings demonstrated that TαPcZn-PDT inhibited the growth of MCF-7 cells in a dose-dependent pattern and had little effect on mammary epithelial MCF-10A cells, indicating that TαPcZn might be an effective and safe photosensitizer for breast cancer.

Growing evidence has shown that phthalocyanine-PDT exert their anti-tumor effects by triggering apoptosis [11,12,13,14] and pyroptosis [28]. Our previous study has proven that TαPcZn-PDT can apparently induce apoptosis in hepatocellular carcinoma Bel-7402 cells and LoVo colon carcinoma cells [26,27]. However, the underlying mechanisms of apoptosis and pyroptosis induced by TαPcZn-PDT in MCF-7 cells remain unclear. Therefore, we quantitated the apoptosis and pyroptosis induced by TαPcZn-PDT in MCF-7 cells. The results showed that TαPcZn-PDT led to an increase in the percentage of apoptotic and pyroptotic cells in a dose-dependent pattern, suggesting that TαPcZn-PDT might induce apoptosis and pyroptosis in MCF-7 cells.

When photosensitizing drugs are excited with an appropriate wavelength of light, ROS are produced, which may result in the photochemical destruction of tumors. Certain studies have reported that ROS play a key role in apoptosis and pyroptosis induced by phthalocyanine-PDT in cancer cells [12,13,14,29]. Our previous study indicated that TαPcZn-PDT induces ROS in LoVo colon carcinoma cells [30]. However, whether ROS are involved in the apoptosis and pyroptosis induced by TαPcZn-PDT in MCF-7 cells remains unknow; therefore, the ROS levels were detected. The results indicated that TαPcZn-PDT dose-dependently induced ROS production, suggesting that ROS might induce apoptosis and pyroptosis in TαPcZn-PDT-treated MCF-7 cells.

Mitochondria are energy-generating organelles that play important roles in various cellular stages, such as adenosine triphosphate production, metabolism, apoptosis and pyroptosis. Mitochondria-targeted phthalocyanine photosensitizers under light irradiation can cause mitochondrial dysfunction, apoptosis and pyroptosis in tumor cells [28,31]. Our previous study has indicated that mitochondria are involved in the apoptosis induced by TαPcZn-PDT in hepatocellular carcinoma Bel-7402 cells and LoVo colon carcinoma cells [26,27]. However, it is not clear whether mitochondria were involved in the apoptosis and pyroptosis induced by TαPcZn-PDT in MCF-7 cells. Therefore, the effect of TαPcZn-PDT on ∆Ψ was examined. The results demonstrated that TαPcZn-PDT induced a decrease in ∆Ψm in MCF-7 cells in a dose-dependent pattern, indicating that mitochondria might modulate the apoptosis and pyroptosis induced by TαPcZn-PDT in MCF-7 cells.

Accumulating evidence has indicated that caspase-1 is a major regulator of pyroptosis [24,25] and caspase-3 is a key protease in apoptosis [22,23] in photosensitizers-PDT-induced cancer. Also, NFκB factors associated with cellular inflammation appear to be a key event in apoptosis and pyroptosis [32,33,34]. Our previous study has shown that caspase-3 is involved in the apoptosis induced by TαPcZn-PDT in hepatocellular carcinoma Bel-7402 cells and LoVo colon carcinoma cells. However, it was not clear whether caspase-3, caspase-1 and NFκB are involved in the apoptosis and pyroptosis induced by TαPcZn-PDT in MCF-7 cells. Therefore, the effects of TαPcZn-PDT on caspase-3, caspase-1 and NFκB were examined. The results indicated that TαPcZn-PDT activated caspase-3, caspase-1 and NFκB in a dose-dependent pattern in MCF-7 cells, indicating that caspase-3, caspase-1 and NFκB might modulate the apoptosis and pyroptosis induced by TαPcZn-PDT in MCF-7 cells.

Certain studies have reported that caspase-1 regulates the apoptosis and pyroptosis of cells [24,25,35,36,37]. However, whether caspase-1 modulates the apoptosis and pyroptosis induced by TαPcZn-PDT in MCF-7 cells has not been ascertained. Therefore, the effect of siRNA-caspase-1 on apoptosis, pyroptosis, caspase-3 and NFκB in TαPcZn-PDT-treated MCF-7 cells was examined. The results revealed that siRNA-caspase-1 attenuated the apoptosis, pyroptosis and activation of caspase-1, caspase-3 and NFκB in the TαPcZn-PDT-treated MCF-7 cells. In summary, we conclude that caspase-1 regulates the phthalocyanine zinc-photodynamic induction of apoptosis and pyroptosis in MCF-7 cells.

## 4. Materials and Methods

### 4.1. Materials

DMEM medium and Opti-MEM medium were purchased from Thermo Fisher scientific (Carlsbad, CA, USA), PBS was purchased from Westang (Shanghai, China), MTT was purchased from Sigma-Aldrich (St. Louis, MO, USA), Calcein AM/PI live/dead cell double-staining kit and reactive oxygen species assay kit were purchased from Solaibao Life Science (Beijing, China), mitochondrial membrane potential assay kit was purchased from Beyotime Technology (Shanghai, China), JetPRIME transfection reagent was purchased from Polyplus (Strasbourg, Bas-Rhin, France), caspase-1 and caspase-3 antibodies were obtained from Cell Signaling Technology (Danvers, MA, USA). NFκB and p-NFκB antibodies were obtained from Santa Cruz(Santa Cruz, CA, USA). Horseradish peroxidase secondary antibodies were purchased from ZSGB-BIO (Beijing, China). MCF-7 breast cancer cells were purchased from Proncell Life Science and Technology (Wuhan, China), and MCF-10A cells were purchased from Otwo Biotech (ShenZhen) Inc. (ShenZhen, China). TαPcZn was constructed as described in our previous report [38].

### 4.2. Cell Culture and Processing

MCF-7 and MCF-10A cells were cultured in DMEM medium containing 10% fetal bovine serum, respectively, in an incubator at 37 °C and 5% CO_2_. Cells at logarithmic growth phase (5 × 10^4^ cells/mL) were inoculated for 24 h. After treatment with the TαPcZn solution at 37 °C in the presence or absence of siRNA-caspase-1, the cells were irradiated with an SS-B instrument (Wuxi Holyglow Physiotherapy Instrument Co., Ltd., Wuxi, China) that emitted red light within a wavelength of 600–700 nm. The light dose was ~53.7 J/cm^2^.

### 4.3. Calcein AM/PI Assay for Cytotoxicity

MCF-7 cells of logarithmic growth phase (5 × 10^4^ cells/mL) were inoculated in culture dishes (20 mm) and grouped in the same cell culture. After the cells were cultured, 1 mL of working solution containing Calcein AM/PI dye was added and incubated for 30 min, washed three times with PBS, and observed under fluorescence microscope. Calcein AM dye penetrates through the living cell membrane into the cell to produce bright green fluorescence, while PI dye does not penetrate the intact living cell membrane, but only the damaged incomplete cell membrane region into the cell and then into the nucleus, producing red fluorescence. Live cell ratio (%) = (live cells number in experimental group/live cells number in blank group) × 100%. The experiment was repeated three times.

### 4.4. MTT Assay for Cell Viability

MCF-7 cells and MCF-10A cells at logarithmic growth stage (5 × 10^4^ cells/mL) were inoculated in 96-well plates and grouped in the same cell culture. After the end of cellular action, 20 uL of MTT solution was added to each well and the culture was continued for 4 h. The medium was removed from each well, 150 uL of DMSO was added to each well, and the cells were incubated in an incubator for 15 min. After the blue-violet crystals were dissolved, the absorbance of each well was measured at 570 nm wavelength with a SpectraMax iD3 multi-function reader (Molecular Devices, Sunnyvale, CA, USA). Cell viability (%) = (optical density value in experimental well − optical density value in blank zeroing well) × 100%/(optical density value in blank control well − optical density value in blank zeroing well). The experiment was repeated 3 times.

### 4.5. DCFH-DA Assay for Cellular ROS Production Levels

Cells of logarithmic growth phase (5 × 10^4^ cells/mL) were inoculated in culture dishes (20 mm) and grouped in the same cell culture. After the cell action, 250 uL of Hoechst 33342 dye was added to each dish and incubated for 30 min; then, 100 uL of pre-mixed DCFH-DA working solution was added to each group and incubated for 30 min. Finally, the cells were washed, and fluorescence intensity was detected by fluorescence microscope. ROS level (%) = (green fluorescence intensity/blue fluorescence intensity) × 100%. The experiment was repeated three times.

### 4.6. JC-1/Hoechst 33342 Assay for Intracellular Mitochondrial Membrane Potential (∆Ψm)

JC-1 is a fluorescent probe widely used to detect the mitochondrial membrane potential of cells. At higher ∆Ψm, JC-1 aggregates in the matrix of mitochondria to form polymers, producing the red fluorescence, while at lower ∆Ψm, JC-1 that cannot aggregate in the matrix of mitochondria is a monomer emitting the green fluorescence. The loss of ∆Ψm was detected by JC-1 assay using fluorescence microscopy. Cells of logarithmic growth phase (5 × 10^4^ cells/mL) were incubated in culture dishes (20 mm) and grouped in the same cell culture. After 500 uL of Hoechst 33342 dye was incubated in each dish for 30 min, 1 mL of pre-mixed JC-1 working solution was incubated for 30 min. Finally, fluorescence intensity was detected by fluorescence microscope. The experiment was repeated three times.

### 4.7. Annexin V/Propidium Iodide (PI) Double-Staining Analysis of Apoptosis and Pyroptosis

Cells at logarithmic growth stage (5 × 10^4^ cells/mL) were inoculated in culture dishes (20 mm) and grouped together in the same cell culture, and the cells were collected at the end of cell action, washed 3 times with PBS, and resuspended with 100 µL buffer, whereby 50 uL buffer mixed with Annexin V-FITC was added first, and then 50 uL buffer mixed with PI was added after 10 min. The cells were incubated for 20 min on ice, protected from light and analyzed using flow cytometry. The experiment was repeated 3 times. The findings are shown as dotplots. In each graph, the percentages of apoptotic cells and pyroptotic cells are indicated in the lower-right and upper-right quadrant, respectively; the *y*-axis corresponds to relative PI staining and the *x*-axis corresponds to the log of the Annexin V-FITC signal.

### 4.8. Western Blot Assay for Protein Expression

Cells of logarithmic growth phase (5 × 10^4^ cells/mL) were inoculated in culture flasks and grouped together in the same cell culture. After the end of cellular action, cells were collected, and 150 µL of cell lysate was added, sonicated and centrifuged at 4 °C for 1 h at 12,000 r/min. Protein concentration was determined by Lowry’s method. An amount of 30 µg of total protein/well was sampled, separated by SDS-PAGE and transferred to PVDF membrane. The samples were closed with 5% skim milk powder for 1 h 30 min at room temperature, incubated with primary antibodies against caspase-1, caspase-3 (1:1000) and NFκB, p-NFκB (1:400), respectively, overnight at 4 °C, rinsed with TBST for 5 min × 3 times, and incubated with 1:5000 horseradish peroxidase-conjugated secondary antibody for 1 h 30 min at room temperature. The gel was first rinsed with TBS for 5 min × 2 times, then rinsed with TBST for 10 min × 1 times, and the ECL was illuminated. Images were acquired by gel imaging image analysis system. Protein level (%) = (protein gray value in experimental group/β-actin protein gray value) × 100%. The experiment was repeated 3 times.

### 4.9. siRNA Transfection

The MCF-7 cells were transfected with siRNAs using transfection jetPRIME reagent according to the manufacturer’s instructions and assayed 48 h after transfection. Negative-control siRNA, positive-control siRNA and against-caspase-1 (12.5 nM) siRNA were all obtained from Thermo Fisher Scientific. The target sequence of the caspase-1 siRNA was forward 5′-CCACUGAAAGAGUGACUUUTT-3′, reverse 5′-AAAGUCACUCUCUUCAGUGGTG-3′ (caspase-1-1) and forward 5′-GGAAGACACUCAUUGAACAUATT-3′, reverse 5′-UAUGUUCAAUGAGUCUUCCAA-3′ (caspase-1-2). The silencing effect was validated by a fluorescence microscope, an inverted microscope and Western blot assay.

### 4.10. Statistical Methods

GraphPad 8.0.2 statistical software was used for analysis. Means between groups were compared by ANOVA, and *p* value less than 0.05 was considered as statistically significant difference.

## 5. Conclusions

In the present study, we found that TαPcZn-PDT resulted in the increase in cytotoxicity and the percentage of apoptotic and pyroptotic cells, the reduction in cell viability and ΔΨm, the production of ROS and the activation of caspase-1, caspase-3 and NFκB in MCF-7 cells. Furthermore, the results also revealed that siRNA-targeting caspase-1 (siRNA-caspase-1) attenuated the effect of TαPcZn-PDT on the apoptosis, pyroptosis and activation of caspase-1, caspase-3 and NFκB in MCF-7 cells. Taken together, we conclude that caspase-1 regulates the apoptosis and pyroptosis induced by TαPcZn-PDT in MCF-7 cells.

## Figures and Tables

**Figure 1 molecules-28-05934-f001:**
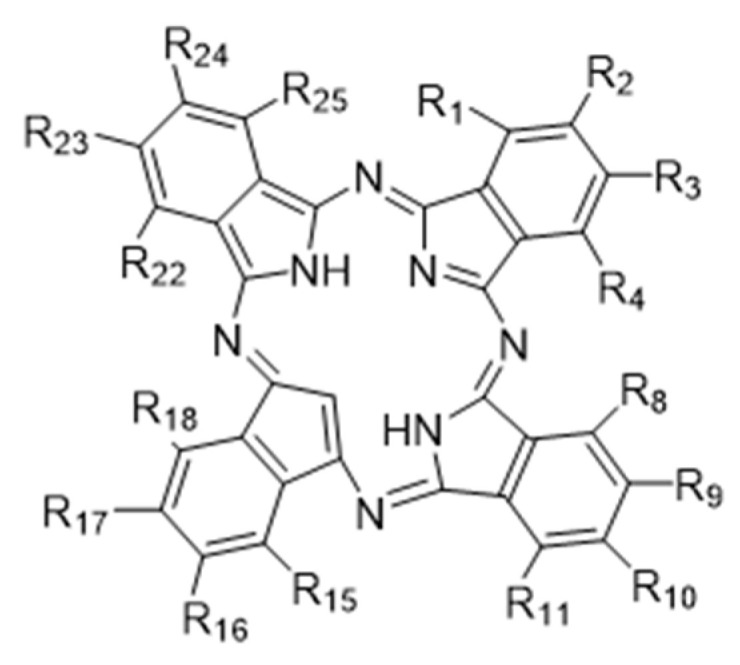
Structure of phthalocyanine.

**Figure 2 molecules-28-05934-f002:**
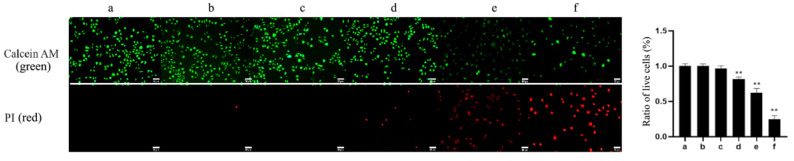
Effect of TαPcZn-PDT on the cytotoxicity of MCF-7 cells. MCF-7 cells were treated with various concentrations of TαPcZn under red light irradiation (53.7 J/cm^2^), and then incubated for 3 h. The cytotoxicity of MCF-7 cells was detected by Calcein AM/PI assay using fluorescence microscopy. Bars under each panel represent 50 μm. Values are the mean of three independent experiments (mean ± standard deviation; ** *p* < 0.01, compared with control treatment). (**a**): Blank control treatment. (**b**): Red light control treatment. (**c**): TαPcZn (0.5 μg/μL) control treatment. (**d**): TαPcZn (0.125 μg/μL)-PDT treatment. (**e**): TαPcZn (0.25 μg/μL)-PDT treatment. (**f**): TαPcZn (0.5 μg/μL)-PDT treatment.

**Figure 3 molecules-28-05934-f003:**
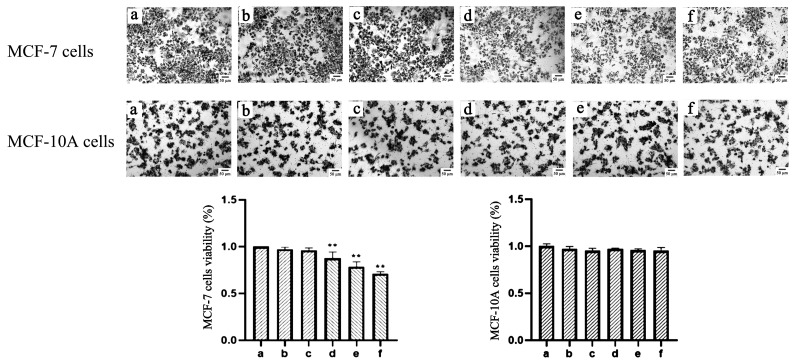
Effect of TαPcZn-PDT on the viability of MCF-7 cells and MCF-10A cells. MCF-7 cells and MCF-10A cells were treated with various concentrations of TαPcZn under red light irradiation (53.7 J/cm^2^), and then incubated for 3 h. The viability of MCF-7 cells and MCF-10A cells was determined via MTT assay using microscopy and multi-function reader. Bars under each panel represent 50 μm. Values are the mean of three independent experiments (mean ± standard deviation; ** *p* < 0.01, compared with control treatment). (**a**): Blank control treatment. (**b**): Red light control treatment. (**c**): TαPcZn (0.5 μg/μL) control treatment. (**d**): TαPcZn (0.125 μg/μL)-PDT treatment. (**e**): TαPcZn (0.25 μg/μL)-PDT treatment. (**f**): TαPcZn (0.5 μg/μL)-PDT treatment.

**Figure 4 molecules-28-05934-f004:**
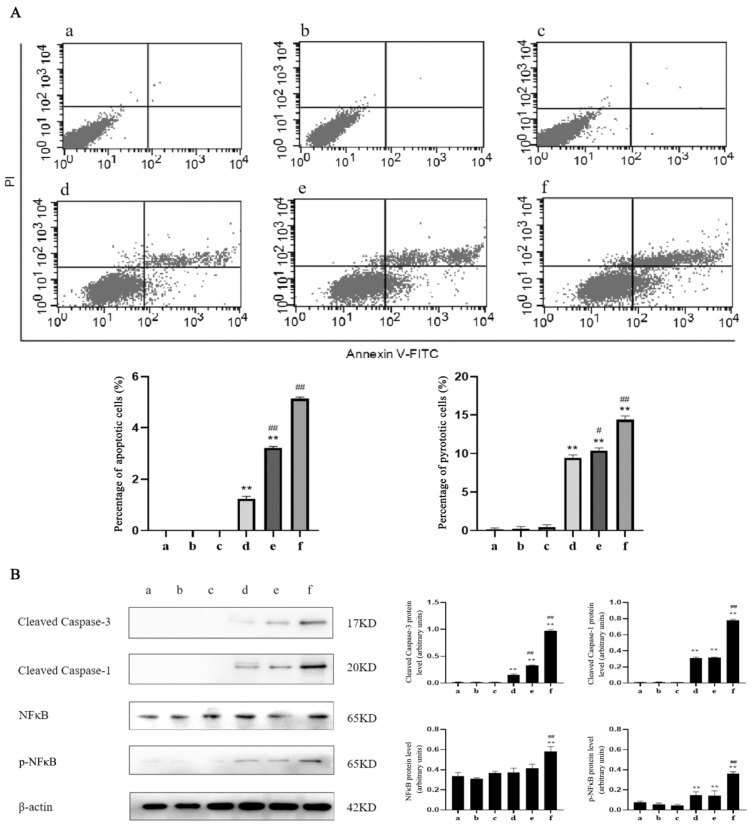
Effect of TαPcZn-PDT on apoptosis and pyroptosis in MCF-7 cells. MCF-7 cells were treated with various concentrations of TαPcZn under red light irradiation (53.7 J/cm^2^), and then incubated for 3 h. The apoptosis and pyroptosis was quantitated by a flow cytometry analysis of Annexin V/PI double-stained cells (**A**), and the expression of caspase-3, caspase-1, NFκB and p-NFκB was analyzed by Western blot assay (**B**). Values are the mean of three independent experiments (mean ± standard deviation; ** *p* < 0.01, compared with control treatment; # *p* < 0.05, compared with TαPcZn (0.125 μg/μL)-PDT treatment; ## *p* < 0.01, compared with TαPcZn (0.125 μg/μL)-PDT treatment). (**a**): Blank control treatment. (**b**): Red light control treatment. (**c**): TαPcZn (0.5 μg/μL) control treatment. (**d**): TαPcZn (0.125 μg/μL)-PDT treatment. (**e**): TαPcZn (0.25 μg/μL)-PDT treatment. (**f**): TαPcZn (0.5 μg/μL)-PDT treatment.

**Figure 5 molecules-28-05934-f005:**
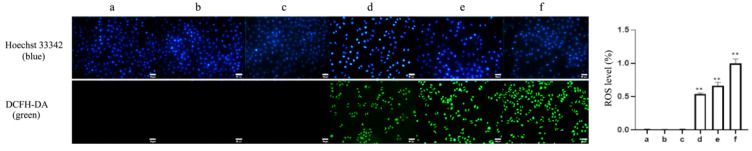
Effect of TαPcZn-PDT on ROS in MCF-7 cells. MCF-7 cells were treated with various concentrations of TαPcZn under red light irradiation (53.7 J/cm^2^), and then incubated for 3 h. Cellular ROS levels were detected by DCFH-DA/Hoechst 33342 assay of fluorescence microscope. Bars under each panel represent 50 μm. Values are the mean of three independent experiments (mean ± standard deviation; ** *p* < 0.01, compared with control treatment). (**a**): Blank control treatment. (**b**): Red light control treatment. (**c**): TαPcZn (0.5 μg/μL) control treatment. (**d**): TαPcZn (0.125 μg/μL)-PDT treatment. (**e**): TαPcZn (0.25 μg/μL)-PDT treatment. (**f**): TαPcZn (0.5 μg/μL)-PDT treatment.

**Figure 6 molecules-28-05934-f006:**
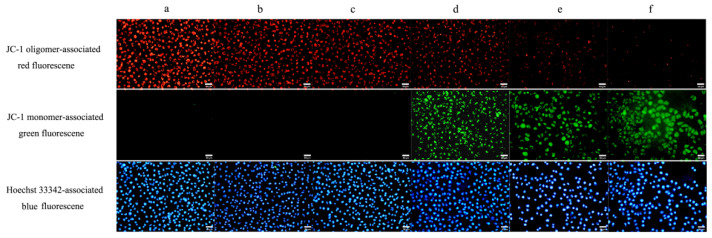
Effect of TαPcZn-PDT on ∆Ψm in MCF-7 cells. MCF-7 cells were treated with various concentrations of TαPcZn under red light irradiation (53.7 J/cm^2^), and then incubated for 3 h. ∆Ψm was detected by JC-1/Hoechst 33342 assay of fluorescence microscope (JCI oligomer-associated red fluorescence; JCI monomer-associated green fluorescence; Hoechst 33342-associated blue fluorescence). Bars under each panel represent 50 μm. (**a**): Blank control treatment. (**b**): Red light control treatment. (**c**): TαPcZn (0.5 μg/μL) control treatment. (**d**): TαPcZn (0.125 μg/μL)-PDT treatment. (**e**): TαPcZn (0.25 μg/μL)-PDT treatment. (**f**): TαPcZn (0.5 μg/μL)-PDT treatment.

**Figure 7 molecules-28-05934-f007:**
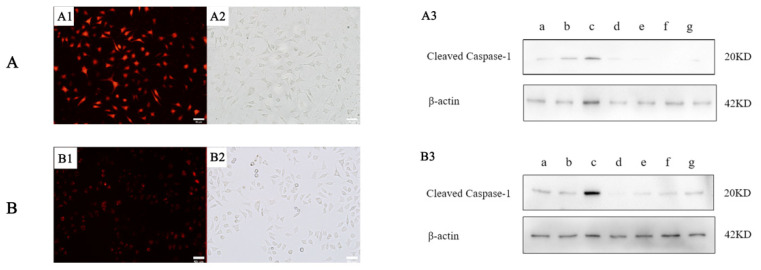
Silencing of caspase-1-1 (**A**) and caspase-1-2 (**B**) in MCF-7 cells. MCF-7 cells were transfected with siRNA-caspase-1-1, siRNA-caspase-1-2, the negative control siRNA and the positive control siRNA using JetPRIME. In the presence of Block-iT Alexa Fluor Red Fluorescent control for 48 h, the MCF-7 cells are exposed to red light irradiation (53.7 J/cm^2^), incubated for 3 h and then, the transfected cells were observed using (**A1**,**B1**) a fluorescence microscope and (**A2**,**B2**) an inverted microscope. Following transfection, the expression levels of caspase-1 protein was detected using (**A3**,**B3**) Western blot assay. Bars under each panel represent 50 μm. (a): Blank control treatment. (b): JetPRIME control treatment. (c): TαPcZn (0.5 μg/μL)-PDT treatment. (d): siRNA-negative control treatment. (e): siRNA-positive control treatment. (f): siRNA-caspase-1 treatment. (g): siRNA-caspase-1/TαPcZn (0.5 μg/μL)-PDT treatment.

**Figure 8 molecules-28-05934-f008:**
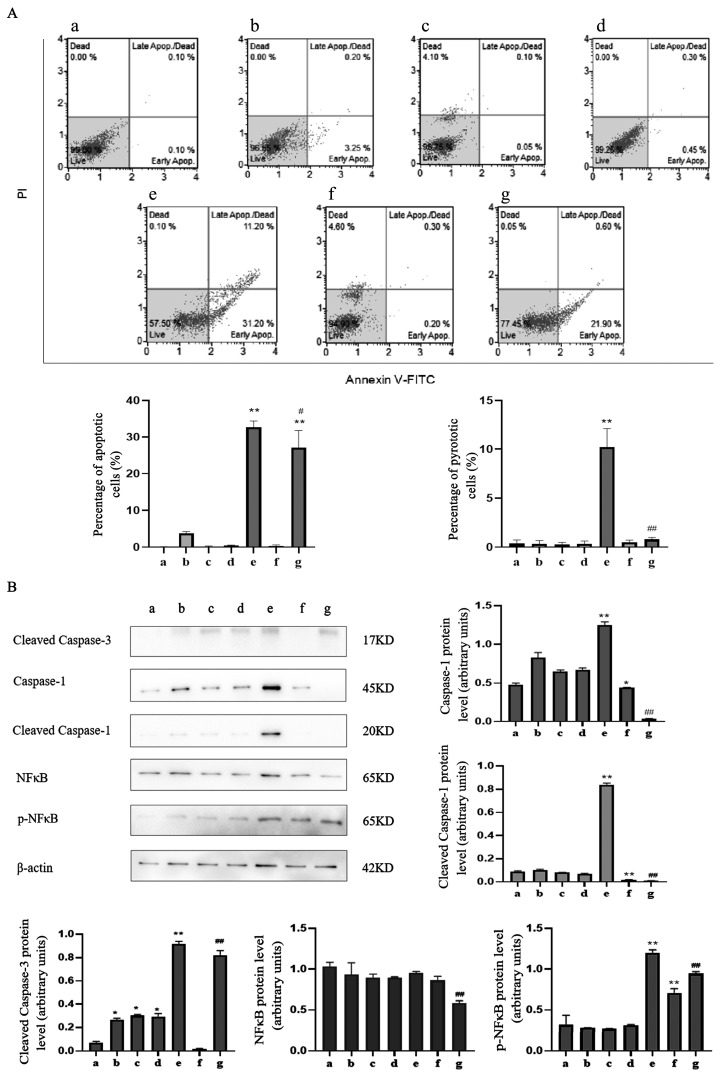
Effect of siRNA-caspase-1 on the apoptosis and pyroptosis induced by TαPcZn-PDT in MCF-7 cells. In the presence of siRNA-caspase-1, MCF-7 cells were treated with 0.5 μg/μL TαPcZn under red light irradiation (53.7 J/cm^2^), and then incubated for 3 h. The apoptosis and pyroptosis was quantitated by the flow cytometry analysis of Annexin V/PI double-stained cells (**A**), and the expression of caspase-3, caspase-1, NFκB and p-NFκB was analyzed by Western blot assay (**B**). Values are the mean of three independent experiments (mean ± standard deviation; * *p* < 0.05, compared with control treatment; ** *p* < 0.01, compared with Blank control treatment; # *p* < 0.05, compared with TαPcZn (0.5 μg/μL)-PDT treatment; ## *p* < 0.01, compared with TαPcZn (0.5 μg/μL)-PDT treatment). (**a**): Blank control treatment; (**b**): siRNA-negative control treatment; (**c**): siRNA-positive control treatment; (**d**): JetPRIME control treatment; (**e**): TαPcZn (0.5 μg/μL)-PDT treatment; (**f**): siRNA-caspase-1 treatment; (**g**): siRNA-caspase-1/TαPcZn (0.5 μg/μL)-PDT treatment.

## Data Availability

The date presented in this study are available from the corresponding author on reasonable request.

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
