# Peer review of "Caspase-1 Regulates the Apoptosis and Pyroptosis Induced by Phthalocyanine Zinc-Mediated Photodynamic Therapy in Breast Cancer MCF-7 Cells"

_molecules, 2023, doi:10.3390/molecules28165934_

Round 1
Reviewer 1 Report
In this paper, the authors found that Caspase-1 regulates the photodynamic induction of apoptosis in MCF-7 cells by phthalocyanine zinc. The discovery is interesting and meaningful. However, I think the authors need to improve the figures and the legends or methods to support the conclusion more clearly.
Fig 1-5 are clearly presented but a bit difficult to get the overview easily. I suggest to show them in a similar way like Fig 6. In addition, the errors bars are not explained in the legend, it is better to show all the data points since the n number is not high, this also applies to other figures.
For Fig 8A, are the NFkB and P-NFkB bands from the same gel, if not, what are the control bands for both gels, this also applies to Fig 11A. It is always better to show the full picture of the western blot to have a fair impression. Fig 8B, how are the values determined? This also applies to Fig 11B.
For Fig 9B, where are the control (reference) bands? What are the meaning of siRNA-NC and siRNA-PC exactly, is there siRNA or any other treatment? if the difference is only with or without siRNA, why there is a strong band for c? Why is this not the same case in Fig 11A? The legend needs to be described in more detail to understand the results.
Why are there two Caspase-1 of different sizes (20KD and 45 KD) in Fig 11A?
Without more detailed information, it is not clear whether the conclusion is convincing or not in its current state as I see.
Reviewer 2 Report
The present study by Ma et al. investigated the photodynamic therapy (PDT) mechanism involving Caspase-1 in breast cancer MCF-7 cells treated with phthalocyanine zinc (TαPcZn). They assessed cytotoxicity, cell viability, mitochondrial membrane potential, ROS production, and cell scorching. Caspase-1, Caspase-3, and P-NFκB were up-regulated in TαPcZn-PDT treated cells, while Caspase-1 silencing decreased scorching and Caspase-3 expression. NFκB factor expression was reduced, and TαPcZn-PDT induced apoptosis in MCF-7 cells but had no significant effect on MCF-10A cells. Therefore, Caspase-1 plays a role in TαPcZn-PDT-induced scorching of breast cancer cells (MCF-7) and can be inhibited by siRNA-Caspase-1 without affecting MCF-10A cells.
This is a very well planned and executed study. However, below are some of my minor comments:
1) Please write a concise conclusion with a subheading to aid readers in quickly grasping the study's findings.
2) If possible, write a detailed caption for Fig. 2A, 2B, and 2C, explaining sub-figures such as Fig. 2Aa, 2Ab, etc..
3) Please include a novelty statement at the end of introduction section.
4) Please minimize the repetition of results in the discussion section and instead focus on exploring potential avenues for obtaining the observed outcomes.
Round 2
Reviewer 1 Report
The authors have sufficiently addressed my concerns.